# Dynamical Persistent Homology via Wasserstein Gradient Flow

## Abstract

In this study, we introduce novel methodologies designed to adapt original data in response to the dynamics of persistence diagrams along Wasserstein gradient flows. Our research focuses on the development of algorithms that translate variations in persistence diagrams back into the data space. This advancement enables direct manipulation of the data, guided by observed changes in persistence diagrams, offering a powerful tool for data analysis and interpretation in the context of topological data analysis.

## 1 Introduction

Persistent homology (Edelsbrunner et al., 2000) analyzes the multi-scale topological features of data, and a persistence diagram summarizes these features by keeping their birth and death across a filtration. Persistence diagram is also the most common topological feature that applied in topological data analyais (TDA) machine learning whose typical extraction can be presented as:

$$\text{Data} \rightarrow \text{Simplicial Complex} \xrightarrow{\text{Dgm}_p \circ F} \text{Persistence Diagram}.$$

Here, we begin with input data, which can be point clouds, graphs, images, or manifolds, and use these to construct abstract simplicial complexes. A filtration, denoted by $F$, creates a sequence of nested simplicial complexes from the data. With this structure, we use a matrix reduction algorithm (Edelsbrunner & Harer, 2022) to compute the persistence diagram, $\text{Dgm}_p$, for the $p$-th homology group. The persistence diagram is a collection of points, each representing the birth and death of a topological feature.

Traditionally, the pipeline flows in one direction: from data to persistence diagrams. However, recent research highlights the need to reverse this process, allowing for filtration or even data adjustments through manipulation of persistence diagrams (Gameiro et al., 2016; Hofer et al., 2020; Oudot & Solomon, 2020; Ballester & Rieck, 2024).

This paper focuses on adapting data based on the dynamics of persistence diagrams along Wasserstein gradient flows. This builds on prior work by using Wasserstein gradient flows to guide the process of modifying data to achieve desired topological characteristics. By developing algorithms that translate changes in persistence diagrams back to the data space, this research enables data analysis and interpretation through TDA.

## 2 Related Work

As mentioned in Introduction 1, persistence diagrams are a powerful tool for summarizing topological information (Edelsbrunner et al., 2000; Edelsbrunner & Harer, 2022; Chazal & Divol, 2018). They have found applications in diverse fields including shape analysis (Poulenard et al., 2018), image analysis (Li et al., 2014; Singh et al., 2007), graph classification (Li et al., 2012; Rieck et al., 2019; Carrière et al., 2020; Wang et al.), link prediction (Yan et al., 2021; Aktas et al., 2019), and various machine learning tasks (Chen et al., 2021a;b; 2024; Wang et al., 2024).

One key aspect of persistence diagrams is their stability with respect to perturbations in the data they are built upon. The most common metric used to compare persistence diagrams is the bottleneck distance, which corresponds to the $\infty$-Wasserstein distance. The stability of persistence diagrams

under the bottleneck distance has been well studied (Cohen-Steiner et al., 2005; Edelsbrunner et al., 2006; Chazal et al., 2009). However, the bottleneck distance can be sensitive to outliers and often provides overly pessimistic bounds, especially in high-dimensional settings. Recent work has shifted focus to the use of $p$-Wasserstein distances, A central challenge in this area has been establishing the stability of persistence diagrams with respect to the $p$-Wasserstein distance(Skraba & Turner, 2020).

The differentiability of persistence diagrams is another important consideration for optimization problems. Early work relied on explicitly computing the derivatives of persistence diagrams with respect to function values, often requiring combinatorial updates (Poulenard et al., 2018). More recently, researchers have leveraged tools from real analytic geometry and o-minimal theory to establish the differentiability of persistence-based functions. Carriere et al. (2021) demonstrated that the persistence map can be viewed as a semi-algebraic map between Euclidean spaces, enabling the definition and computation of gradients for a wide range of persistence-based functions. Moreover, Leygonie (2022); Leygonie et al. (2022) not only provide a theoretical foundation for the application of gradient-based optimization techniques to persistence-based problems, but also establish a framework for understanding how changes in data parameters impact persistence diagrams.

Several previous studies have explored methods for manipulating data through modifications to persistence diagrams. Gameiro et al. (2016) developed a method for point cloud continuation by using the Newton-Raphson method to move a persistence diagram towards a target diagram, iteratively updating the point cloud to achieve the desired topological features. Poulenard et al. (2018) presented an approach for optimizing real-valued functions based on topological criteria, using derivatives of persistence diagrams to guide the modification of the function and achieve desired topological properties. Corcoran & Deng (2020) focused on regularizing the computation of persistence diagram gradients to improve the optimization process in data science applications.

These works demonstrate the potential of using persistence diagrams as a tool for manipulating data to achieve desired topological characteristics. The current work aims to build upon these efforts by developing novel algorithms that translate variations in persistence diagrams back into the data space, with a focus on the $p$-Wasserstein distance and gradient-based optimization techniques.

## 3 Preliminaries

This study's theoretical foundation rests on two key areas: Wasserstein gradient flows (WGFs) and differentiable persistence diagrams. In this Preliminaries section, we first introduce the concept of gradient flows in Hilbert spaces, providing the necessary mathematical background. We then extend this to gradient flows in Wasserstein spaces, a crucial framework for our analysis. Finally, we present the fundamentals of differentiable persistent homology, an essential tool in topological data analysis that connects persistent homology with differentiable optimization techniques. These concepts form the basis for the methodologies and analyses presented in subsequent chapters.

### 3.1 Gradient Flow in Hilbert Space

Gradient flow in Hilbert space (Ambrosio et al., 2008) is a fundamental concept in the analysis of variational problems and partial differential equations. It describes the evolution of a function in a Hilbert space under the influence of its gradient, leading to a minimization of the associated energy functional.

Let $\mathcal{H}$ be a Hilbert space with inner product $\langle \cdot, \cdot \rangle$ and norm $\| \cdot \|$. Consider an energy functional $E : \mathcal{H} \to \mathbb{R}$, which is typically assumed to be Fréchet differentiable. The gradient flow of $E$ is the solution to the differential equation

$$\begin{cases} \dfrac{du(t)}{dt} = -\nabla E(u(t)) & \text{for } t > 0, \\ u(0) = u_0 \end{cases} \tag{1}$$

where $u(t) \in \mathcal{H}$ represents the state of the system at time $t$, and $\nabla E(u)$ denotes the gradient of $E$ at $u$.

The gradient $\nabla E(u)$ is defined by the property that for all $v \in \mathcal{H}$,

$$\frac{d}{d\epsilon} E(u + \epsilon v)\bigg|_{\epsilon=0} = \langle \nabla E(u), v \rangle. \tag{2}$$

This ensures that the direction of the gradient $\nabla E(u)$ is the direction of the steepest ascent of the functional $E$. Consequently, the negative gradient $-\nabla E(u)$ points in the direction of the steepest descent, which is why it appears in Equation equation 1.

One of the key properties of gradient flow is that it decreases the energy functional over time. Specifically, if $u(t)$ is a solution to Equation equation 1, then

$$\frac{d}{dt} E(u(t)) = \left\langle \nabla E(u(t)), \frac{du(t)}{dt} \right\rangle = -\|\nabla E(u(t))\|^2 \leq 0. \tag{3}$$

This implies that $E(u(t))$ is non-increasing along the curve $u(t)$. Moreover, since $\frac{d}{dt} E(u(t)) = 0 \Leftrightarrow \|\nabla E(u(t))\| = 0$, if $E$ has a unique stationary point $u^\star$, $u(t)$ converges to it as $t \to \infty$ under appropriate conditions.

## 3.2 PROXIMAL MAP IN HILBERT SPACE

From a computational perspective, discretization schemes are essential for approximating gradient flows in Hilbert spaces. The implicit Euler scheme, also known as the minimizing movement scheme or proximal map (Bubeck et al., 2015), is particularly effective for this purpose. This scheme approximates the continuous-time gradient flow by a sequence of discrete minimization problems:

$$u_{k+1} = \arg\min_{v \in \mathcal{H}} \left( \frac{1}{2\tau} \|u_k - v\|^2 + E(v) \right) \tag{4}$$

where $\tau > 0$ is the time step, $u_k$ represents the approximation at the $k$-th time step, $E$ is the energy functional, and $\mathcal{H}$ is the Hilbert space. This scheme not only provides a practical method for numerical computations but also offers insights into the theoretical properties of gradient flows, such as their connection to proximal operators and convex analysis.

## 3.3 GRADIENT FLOW IN WASSERSTEIN SPACE

Gradient flow in Wasserstein space extends the concept of gradient flow from Hilbert spaces to the space of probability measures. This framework is particularly valuable for studying partial differential equations and variational problems involving mass transport. While this subsection provides a brief introduction, readers are encouraged to consult Ambrosio et al. (2008) for a more comprehensive treatment.

We work in $(\mathcal{P}_2(X), W_2)$; formal definitions of $\mathcal{P}_p(X)$ and $W_p(\mu, \nu)$ are provided in Appendix A.

With a series of works by Benamou & Brenier (2000); Otto (2001); Ambrosio et al. (2008), the concept of gradient flow in Wasserstein space has been rigorously established.

Consider an energy functional $J : \mathcal{P}_2(\mathbb{R}^d) \to \mathbb{R}$ defined on a Wasserstein space, where $\mathcal{P}_2(\mathbb{R}^d)$ denotes the space of probability measures with finite second moments as in Equation equation 13. A curve $(\mu_t)_{t \geq 0}$ of probability measures is called a Wasserstein gradient flow for the functional $J$ if it satisfies the following continuity equation in a weak sense:

$$\frac{\partial \mu_t}{\partial t} = \nabla \cdot \left( \mu_t \nabla \frac{\delta J}{\delta \mu}(\mu_t) \right). \tag{5}$$

Here, $\frac{\delta J}{\delta \mu} : \mathbb{R}^d \to \mathbb{R}$ represents the first variation of $J$ at $\mu$, defined by

$$\frac{d}{d\varepsilon} J(\mu + \varepsilon \xi)\bigg|_{\varepsilon=0} = \int \frac{\delta J}{\delta \mu} d\xi = \left\langle \frac{\delta J}{\delta \mu}, \xi \right\rangle, \quad \text{for all } \xi \text{ with } \int d\xi = 0.$$

This first variation can be interpreted as the "functional derivative" of $J$ with respect to $\mu$, extending the concept of derivatives to measure spaces. Consequently, $\nabla \frac{\delta J}{\delta \mu}(\mu_t) : \mathbb{R}^d \to \mathbb{R}^d$ for $t \geq 0$

represents a time-dependent family of vector fields. Chewi et al. (2020) refer to this as the Wasserstein gradient, denoted as $\nabla_{W_2}(\mu_t) := \nabla \frac{\delta J}{\delta \mu}(\mu_t)$. This term draws an analogy with the standard gradient in Euclidean spaces due to its similar role in describing the flow of probability measures. Additionally, some literature defines the Wasserstein gradient as $\nabla_{W_2}(\mu_t) := -\nabla \cdot \left( \mu_t \nabla \frac{\delta J}{\delta \mu}(\mu_t) \right)$ to maintain consistency with Equation equation 1.

### 3.4 JKO Scheme

The JKO scheme, named after Jordan, Kinderlehrer, and Otto, is a time-discretization method for approximating the Wasserstein gradient flow (Jordan et al., 1998), It involves solving a sequence of minimization problems:

$$\mu_{k+1} = \arg\min_{\mu} \left( \frac{1}{2\tau} W_2^2(\mu, \mu_k) + J(\mu) \right),$$

where $\tau$ is the time step. The transition from the continuous Wasserstein gradient flow to the discrete JKO scheme can be understood through the concept of minimizing movement which approximates the continuous flow by discrete steps. The convergence of the JKO scheme to the continuous Wasserstein gradient flow as $\tau \to 0$ is established through $\Gamma$-convergence (Ambrosio et al., 2008). The functional $J(\mu)$ often represents the internal energy or potential energy of the system, and its gradient with respect to the Wasserstein metric drives the evolution of the distribution (Villani et al., 2009; Villani, 2021). The JKO scheme is particularly useful in numerical simulations of diffusion processes and other phenomena described by partial differential equations (PDEs) in the Wasserstein space (Santambrogio, 2015). Applications of the JKO scheme include image processing, machine learning, and fluid dynamics, where the evolution of distributions is of interest (Peyré et al., 2019).

### 3.5 Differentiability on Persistence Diagrams

Nigmetov & Morozov (2024) introduces a novel method that directly uses the cycles and chains involved in the persistence computation to define gradients for larger subsets of the data. This contrasts with prior methods that focused solely on individual PD points. The authors also demonstrate that this approach can significantly reduce the number of steps required for optimization. Our work builds upon this idea, allowing it to adapt data based on variations in PDs along Wasserstein gradient flows.

## 4 Methodology

We introduce our basic settings as follows. Let a function $f : \mathcal{K} \to \mathbb{R}$, where $\mathcal{K}$ denotes a simplicial complex, induce a filtration. This function assigns a real value to each simplex in the complex, thereby defining a sequence of subcomplexes $\mathcal{K}(t) = \{\sigma \in \mathcal{K} \mid f(\sigma) \le t\}$ for $t \in \mathbb{R}$. Let $\mathrm{dgm}_p(\mathcal{K}, f)$ be the algorithm to compute $p$-th persistence homology based on Nigmetov & Morozov (2024, Algorithm 1). In this study, we don't take essential features into account. To construct the dynamical persistence diagram, we consider two situations: with or without a target persistence diagram.

### 4.1 Dynamical Persistent Homology via McCann Interpolation

In this subsection, we assume that the target persistence diagram is known *a priori*. Our objective is to develop a learnable process for generating persistence diagrams guided by a dynamical system. To approach this problem, we draw upon two fundamental theorems from optimal transport theory: the Benamou-Brenier Theorem and the McCann Interpolation Theorem. The Benamou-Brenier Theorem provides a dynamic formulation of optimal transport, allowing us to view the transport problem as a fluid flow minimizing a kinetic energy functional. Meanwhile, the McCann Interpolation Theorem offers a way to construct geodesics in the space of probability measures, which is crucial for understanding the geometry of persistence diagrams. By leveraging these theorems, we can formulate a continuous evolution of persistence diagrams that converges to the target diagram while respecting the underlying topological structure of the data.

The Benamou-Brenier theorem shows that the optimal transport cost can be expressed as the infimum of the action integral over all possible time-dependent probability measures $\mu_t$ and velocity fields $v_t$ that transport $\mu_0$ to $\mu_1$.

**Theorem 1 (Benamou-Brenier (Benamou & Brenier, 2000))** *Let $\mu_0$ and $\mu_1$ be two probability measures on $\mathbb{R}^d$ with finite second moments. The squared Wasserstein distance $W_2^2(\mu_0, \mu_1)$ between $\mu_0$ and $\mu_1$ can be expressed as:*

$$W_2^2(\mu_0, \mu_1) = \inf_{(\mu_t, v_t)} \int_0^1 \int_{\mathbb{R}^d} \|v_t(x)\|^2 \, d\mu_t(x) \, dt \tag{6}$$

*subject to the continuity constraints:*

$$\frac{\partial \mu_t}{\partial t} + \nabla \cdot (\mu_t v_t) = 0. \tag{7}$$

On the other hand, McCann's interpolation theorem describes the displacement interpolation between two probability measures in the context of optimal transport, showing that the interpolation is a geodesic in the Wasserstein space.

**Theorem 2 (McCann Interpolation (Villani, 2021))** *Let $\mu_0$ and $\mu_1$ be two probability measures on $\mathbb{R}^d$ with finite second moments, and let $T$ be the optimal transport map pushing $\mu_0$ forward to $\mu_1$. Define the displacement interpolation $\mu_t$ for $t \in [0, 1]$ by:*

$$\mu_t = ((1-t)\,\mathrm{Id} + tT)\#\mu_0, \tag{8}$$

*where $\#$ denotes the push-forward operation, $\mathrm{Id}$ is the identity map and $(1-t)\,\mathrm{Id} + tT$ is the interpolation map. Then:*

1. *$\mu_t$ is a probability measure for each $t \in [0, 1]$,*

2. *The curve $\{\mu_t\}_{t \in [0,1]}$ is a geodesic in the Wasserstein space $(\mathcal{P}_2(\mathbb{R}^d), W_2)$,*

3. *The Wasserstein distance between $\mu_0$ and $\mu_1$ can be expressed as:*

$$W_2^2(\mu_0, \mu_1) = \int_{\mathbb{R}^d} \|x - T(x)\|^2 \, d\mu_0(x). \tag{9}$$

Therefore, we have the following remark.

**Remark 3 (McCann Interpolation is a WGF (Ambrosio et al., 2008))** *Let $T_t = (1-t)\,\mathrm{Id} + tT$. Then, $(\mu_t, v_t)$ given by McCann interpolation*

$$\mu_t = T_t \# \mu_0 \tag{10}$$

$$v_t(x) = \begin{cases} T(x_0) - x_0 & \text{if } x = T_t(x_0), \text{where } x_0 \text{ is the initial position to } x. \\ 0 & \text{otherwise} \end{cases} \tag{11}$$

*satisfies Equations equation 6 and equation 7 in Theorem 1.*

The McCann interpolation can be viewed as a specific instance of a dynamical process. Based on this insight, we have developed an algorithm that leverages McCann interpolation to guide the optimization within the context of persistence diagrams.

In the Algorithm 1, the $\mathrm{OptimalTransportPlan}$ can be efficiently computed using the Sinkhorn algorithm (Cuturi, 2013), known for its rapid convergence and computational efficiency. Alternatively, conventional linear programming methods can also be employed. The optimal transportation plan $\pi$ is represented as an $n \times m$ matrix, where $n$ and $m$ denote the cardinalities of the source persistence diagram $X^{(k)}$ and the target persistence diagram $Z$, respectively. Each entry $\pi_{ij}$ in the matrix indicates the amount of mass to be transported from $x_i \in X^{(k)}$ to $z_j \in Z$. The mass associated with each point $x_i \in X^{(k)}$ and $z_j \in Z$ is $1/n$ and $1/m$, respectively.

In cases where $n$ and $m$ are not equal, the $\mathrm{TargetDgm}$ function computes an appropriate target persistence diagram. This is achieved by using the barycenter to determine the target locations, calculated as follows:

$$x_i = \frac{\sum_{j=1}^m \pi_{ij} z_j}{\sum_{j=1}^m \pi_{ij}} \quad \text{for all } i. \tag{12}$$

---

**Algorithm 1** Dynamical Persistent Homology via McCann Interpolation

---

1: **Input:** $\mathcal{K} = \{\sigma_i\}_{i=1}^l$ the simplices, $f^{(1)} : \mathcal{K} \mapsto \mathbb{R}$ the initial filtration function, $K$ the total number of step for the dynamic process, $Z$ the target persistence diagram, $S$ the number of steps to perform filtration updates.
2: **for** $k \leftarrow 1$ to $K$ **do**
3:     $t \leftarrow \frac{1}{K-k+1}$
4:     $X^{(k)} \leftarrow \mathrm{dgm}_p(\mathcal{K}, f^{(k)}) = \{x_i^{(k)}\}_{i=1}^n$               ▷ Compute persistence diagram
5:     $\pi \leftarrow \mathrm{OptimalTransportPlan}(X^{(k)}, Z)$      ▷ Use Sinkhorn Algorithm (Cuturi, 2013)
6:     $X_1 \leftarrow \mathrm{TargetDgm}(\pi)$                          ▷ Use Equation equation 12
7:     $Y^{(k)} \leftarrow (1-t)X^{(k)} + tX_1$                 ▷ Use McCann Interpolation
8:     $f \leftarrow f^{(k)}$
9:     **for** $s \leftarrow 1$ to $S$ **do**
10:         $\mathcal{L} \leftarrow \mathrm{Loss}(\mathrm{dgm}_p(\mathcal{K}, f), Y^{(k)})$
11:         $\nabla f \leftarrow \mathrm{CriticalSetMethod}(\mathcal{L}, f)$      ▷ (Nigmetov & Morozov, 2024, Algorithm 3)
12:         $f \leftarrow f - \eta \nabla f$
13:     $f^{(k+1)} \leftarrow f$
    **return** $\{X^{(k)}\}_{k=1}^K, \{Y^{(k)}\}_{k=1}^K, \{f^{(k)}\}_{k=1}^K$

---

The Algorithm 1 demonstrates the use of McCann interpolation to guide optimization on persistence diagrams. The core idea is to obtain the next persistence diagram through McCann interpolation and employ a learnable scheme, specifically the critical set method (Nigmetov & Morozov, 2024), to determine the subsequent filtration. This approach allows for the construction of the entire dynamical process of the persistence diagram.

The results $X^{(k)}, Y^{(k)}, f^{(k)}$ represent the persistence diagram, the next target persistence diagram, and the filtration at step $k$, respectively. We explicitly store $Y^{(k)}$ to maintain the next step's persistence diagram because the learnable scheme cannot guarantee that the persistence diagram is always achievable (Nigmetov & Morozov, 2024). This is also why we do not compute the full trajectories before learning the filtration. Instead, we compute the McCann interpolation at each step to ensure that the persistence diagram progresses towards the target persistence diagram. In other words, this algorithm optimizes the filtration towards the final target rather than intermediate states.

The proposed algorithm is particularly well-suited for applications where a target persistence diagram is explicitly defined. Additionally, in the context of McCann interpolation, our algorithm guarantees that the measure $\mu_t$ evolves along the geodesic in Wasserstein space, provided that the target persistence diagram is achievable at each step. This property ensures that the topological features of the data, as captured by persistence diagrams, change towards the target diagram in the best possible manner.

## 4.2 Dynamical Persistent Homology via Energy Functional

The previous subsection demonstrated the dynamical persistence diagrams along McCann interpolation trajectories, given a target persistence diagram. In this subsection, we extend our methodology to scenarios without a predefined target diagram. As introduced in Subsection 3.3, a dynamical system describes the temporal evolution of a state according to a set of rules or equations. In the context of Wasserstein gradient flow, the probability measure $\mu_t$ evolves following the gradient flow of an energy functional $J$ in the Wasserstein space. This evolution follows a path that locally minimizes $J$ most efficiently with respect to the Wasserstein distance. The choice of functional $J$ determines the system's ultimate configuration, allowing for diverse outcomes depending on the specific energy landscape.

To perform the dynamical persistence diagram through Wasserstein gradient flows, we propose the following Algorithm 2.

Algorithm 2, akin to Algorithm 1, outputs $X^{(k)}$, $Y^{(k)}$, and $f^{(k)}$, which represent the current persistence diagram, the target persistence diagram, and the filtration at step $k$, respectively. The key distinction lies in Algorithm 2's utilization of the JKO scheme to compute the target persistence

---

**Algorithm 2** Dynamical Persistent Homology via Energy Functional

---

1: **Input:** $\mathcal{K} = \{\sigma_i\}_{i=1}^l$ the simplices, $f^{(1)} : \mathcal{K} \mapsto \mathbb{R}$ the initial filtration function, $K$ the total number of step for the dynamic process, $S$ the number of steps to perform filtration updates, $\tau$ the step size in JKO, $J$ the energy functional.
2: **for** $k \leftarrow 1$ to $K$ **do**
3:   $X^{(k)} \leftarrow \mathrm{dgm}_p(\mathcal{K}, f^{(k)}) = \{x_i^{(k)}\}_{i=1}^n$        ▷ Compute persistence diagram
4:   $\left(y_1^{(k)}, y_2^{(k)}, \dots, y_n^{(k)}\right) \leftarrow \left(x_1^{(k)}, x_2^{(k)}, \dots, x_n^{(k)}\right)$
5:   $\mu_k \leftarrow \frac{1}{n} \sum_{i=1}^n \delta_{x_i^{(k)}}$
6:   $\mu \leftarrow \frac{1}{n} \sum_{i=1}^n \delta_{y_i^{(k)}}$             ▷ Initialize a new measure
7:   $\mu \leftarrow \arg\min_\mu \frac{1}{2\tau} W_2^2(\mu_k, \mu) + J(\mu)$         ▷ Perform JKO
8:   $Y^{(k)} \leftarrow \{y_i^{(k)}\}_{i=1}^n$             ▷ The target at step $k$
9:   $f \leftarrow f^{(k)}$
10:   **for** $s \leftarrow 1$ to $S$ **do**
11:     $\mathcal{L} \leftarrow \mathrm{Loss}(\mathrm{dgm}_p(\mathcal{K}, f), Y^{(k)})$
12:     $\nabla f \leftarrow \mathrm{CriticalSetMethod}(\mathcal{L}, f)$    ▷ (Nigmetov & Morozov, 2024, Algorithm 3)
13:     $f \leftarrow f - \eta \nabla f$
14:   $f^{(k+1)} \leftarrow f$
  **return** $\{X^{(k)}\}_{k=1}^K, \{Y^{(k)}\}_{k=1}^K, \{f^{(k)}\}_{k=1}^K$

---

diagram at each iteration. Importantly, we initialize the target persistence diagram with the current persistence diagram, ensuring equal cardinalities. This approach enables us to leverage the energy functional to guide the optimization process without prior knowledge of the final target persistence diagram.

## 5   CASE ILLUSTRATIONS

The most beneficial aspect of the proposed methodology is its ability to capture the dynamic evolution of topological features in complex data. To demonstrate the effectiveness of our approach, inspired by the examples in Carriere et al. (2021), we present two case illustrations: circle denoising and circle emerging. These examples showcase the power of dynamical persistence diagrams in uncovering the underlying topological structures of noisy data and emerging patterns. In both cases, we use Rips filtration.

### 5.1   CIRCLE DENOISING

Our first illustration concerns the denoising of a circle. The input data consists of a set of points in a 2D plane, sampled from a circle with added Gaussian noise. This noisy data forms a perturbed circle (see Appendix B, Fig. 4). The objective is to remove the noise and highlight the primary 1-dimensional topological feature, namely the circle itself.

### 5.1.1   REPULSION LOSS

In this experiment, we need to prevent the points from clustering together. To achieve this, we introduce an auxiliary loss function designed to enforce point separation. Specifically, given the point set $\mathcal{K}_0 = \{\sigma_i\}_{i=1}^n$, where each $\sigma_i \in \mathbb{R}^2$, the repulsion loss is defined as follows:

$$\mathrm{loss} = \sum_{i=1}^n \sum_{j=1, j\neq i}^n \frac{1}{\|\sigma_i - \sigma_j\|^2 + \epsilon}$$

This formula calculates the repulsion loss for the point set $\mathcal{K}_0$. The parameter $\epsilon$ is a small positive constant that ensures the denominator does not become zero, thereby avoiding numerical instability.

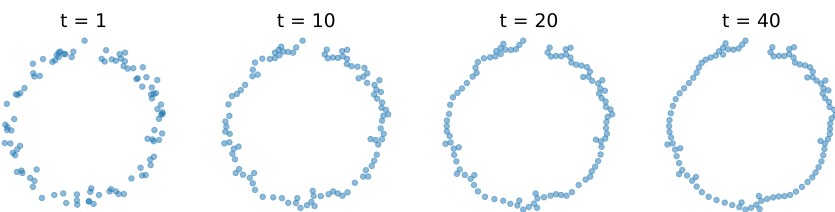

Figure 1: The evolution of the noisy circle data towards the target circle persistence diagram using McCann interpolation on the 0th persistence diagram and a denoising algorithm (Nigmetov & Morozov, 2024) on the 1st persistence diagram.

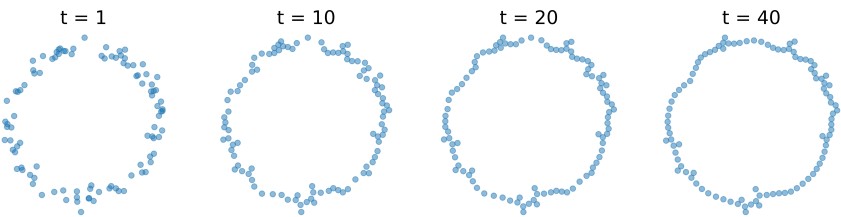

Figure 2: The evolution of the noisy circle data towards the target circle persistence diagram using McCann interpolation on the 0th persistence diagram and energy functional on the 1st persistence diagram.

### 5.1.2 CIRCLE DENOISING VIA MCCANN INTERPOLATION ON 0TH PERSISTENCE DIAGRAM

The target persistence diagram represents a 1-dimensional circle, which is the primary topological feature of interest. We apply Algorithm 1 to denoise the 0th persistence diagram, using the target persistence diagram as a reference. More specifically, we set the target of the 0th persistence diagram to $(0, 0.05)$ in the experiment, aiming to give these points a reasonable distance from nearby points. For the 1st persistence diagram, we employ a built-in denoising method. The McCann stage is shown in Figure 1, and the overall evolution is summarized in Figure 2.

See Appendix B (Figs. 5, 6, 7, and 8) for the 0th and 1st PD-evolution panels that corroborate these trends.

However, we observe a gap at the top of the circle in the results. This occurs because the McCann interpolation is applied solely to the 0th persistence diagram, neglecting the 1st persistence diagram. The built-in denoising process lacks control over the upper left 1st persistence pair, which represents the circle. To address this issue, we apply Algorithm 2 to denoise the 1st persistence diagram and let the persistence pair move towards the left.

### 5.1.3 CIRCLE DENOISING VIA ENERGY FUNCTIONAL ON 1ST PERSISTENCE DIAGRAM

To achieve the goal we mentioned above, we apply Algorithm 2 with the energy functional

$$J(\mu) = \frac{1}{2} \mathbb{E}_{(x,y)\sim\mu} \left[ \min \left( x^2 + (y - 1.2)^2, \frac{(x-y)^2}{2} \right) \right].$$

It makes the points near to the diagonal move towards the diagonal, while points far from the diagonal move towards $(0, 1.2)$. The results of this process are illustrated in Figure 2; see Appendix B (Figs. 5, 6, 7, and 8) for PD-evolution panels. In JKO, we use the sliced Wasserstein implementation from Bonet et al. (2022) for efficiency purposes. As shown, the algorithm not only effectively removes noise from the data but also fills gaps at the top of the circle successfully.

### 5.2 CIRCLE EMERGING

Given a set of random 2D points on a plane, we observe that the first persistence diagram is not always empty. In this case, the objective is to enhance the circles represented by the first persistence

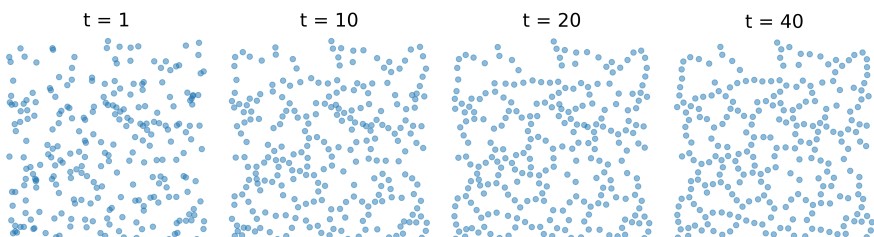

Figure 3: The evolution of the random 2D points towards the target circle persistence diagram using McCann interpolation on the 0th persistence diagram and energy functional on the 1st persistence diagram.

diagram. In the experiment, we independently and identically sample 250 points from the uniform distribution on the square $[-1, 1] \times [-1, 1]$. Similar to the previous case, we apply the McCann algorithm with a target of $(0, 0.08)$ on the 0th persistence diagrams. Additionally, we apply Algorithm 2 with the functional

$$J(\mu) = \frac{1}{4}\mathbb{E}_{(x,y)\sim\mu}\left[(y - (x + 0.15))^2 + x^2\right]$$

to encourage the first persistence pairs above a certain threshold to evolve away from the diagonal line and drift to the left. The results of this process are illustrated in Figure 3. PD-evolution panels are provided in Appendix B (Figs. 9 and 10).

## 6 LIMITATIONS AND FURTHER WORKS

First, the current learnable persistence diagram scheme cannot be executed on a GPU, which hinders the scalability of the proposed algorithms when applied to large datasets. This limitation restricts the practical application of these methods in real-world scenarios where computational efficiency is crucial. Second, the proposed methods exclusively consider persistence diagrams. It is worth exploring whether other topological features, such as Zigzag persistence or multiparameter persistence, can be integrated into the proposed framework to enhance its versatility. Third, there is a need to establish statistical theories for the proposed methods when utilizing different energy functionals. For instance, when deriving dynamical persistence diagrams from a time series of real-world data, it is essential to determine the statistical properties of these diagrams. Additionally, identifying an appropriate energy functional that governs the generation of dynamical persistence diagrams from the time series is crucial. Furthermore, guidelines for selecting suitable energy functionals for various tasks should be developed to ensure optimal performance. Finally, a more effective approach is required to address the conflicts arising from singleton losses (Nigmetov & Morozov, 2024) in the proposed methods, as these conflicts can significantly impact the accuracy and reliability of the results.

## 7 CONCLUSION

In this study, we propose two methods to integrate Wasserstein gradient flow into the learning process for persistence diagrams. First, by leveraging McCann interpolation, we develop an algorithm that optimizes persistence diagrams along the geodesic in the Wasserstein space. This approach ensures that the optimization path respects the intrinsic geometry of the space, leading to more accurate and meaningful results. Second, we introduce an energy functional-based approach that guides the optimization process without requiring a predefined target persistence diagram. This method allows for greater flexibility and adaptability in various applications. We demonstrate the effectiveness of our methods through case studies, including the denoising of a circle and the emergence of a circle from noisy data. These examples illustrate how our techniques can enhance the temporal evolution of topological features, highlighting the potential of dynamical persistence diagrams in topological data analysis.

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

## A    BACKGROUND ON WASSERSTEIN SPACE

The Wasserstein space is the space of probability measures with finite $p$-th moments, equipped with the Wasserstein distance. More formally, let $(X, d)$ be a metric space. The Wasserstein space of order $p$ over $(X, d)$, denoted by $\mathcal{P}_p(X)$, is the set of all probability measures $\mu$ on $X$ with finite $p$-th moment:

$$\mathcal{P}_p(X) = \left\{ \mu \in \mathcal{P}(X) \mid \int_X d(x, x_0)^p \, d\mu(x) < \infty \right\} \tag{13}$$

where $x_0$ is a fixed reference point in $X$. The Wasserstein distance of order $p$ between two probability measures $\mu, \nu \in \mathcal{P}_p(X)$ is defined as

$$W_p(\mu, \nu) = \left( \inf_{\pi \in \Pi(\mu, \nu)} \int_{X \times X} d(x, y)^p \, d\pi(x, y) \right)^{\frac{1}{p}},$$

where $\Pi(\mu, \nu)$ is the set of all couplings of $\mu$ and $\nu$.

## B    ADDITIONAL FIGURES

### B.1    CIRCLE DENOISING

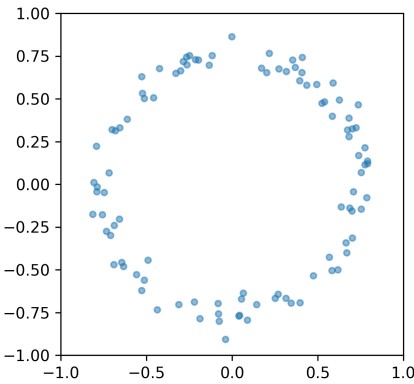

Figure 4: Noisy circle input: points sampled from a circle with Gaussian noise.

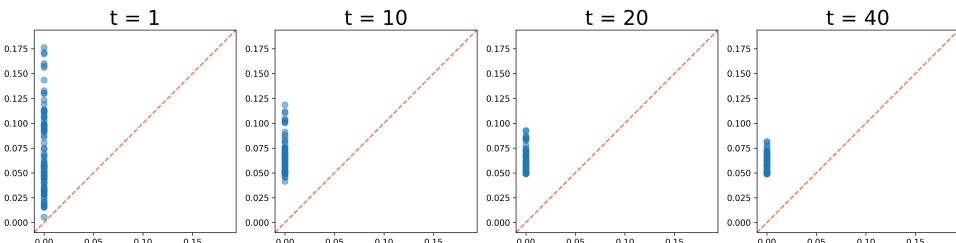

Figure 5: Circle denoising — 0th persistence diagram evolution (McCann stage).

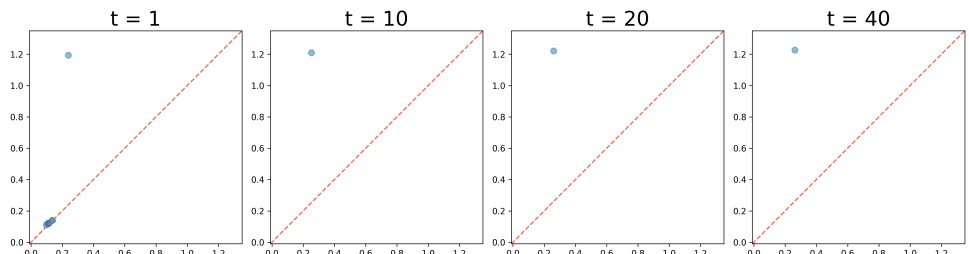

Figure 6: Circle denoising — 1st persistence diagram evolution (McCann stage).

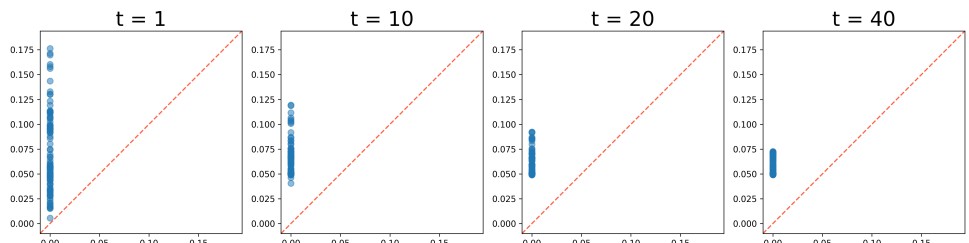

Figure 7: Circle denoising — 0th persistence diagram evolution (combined stages).

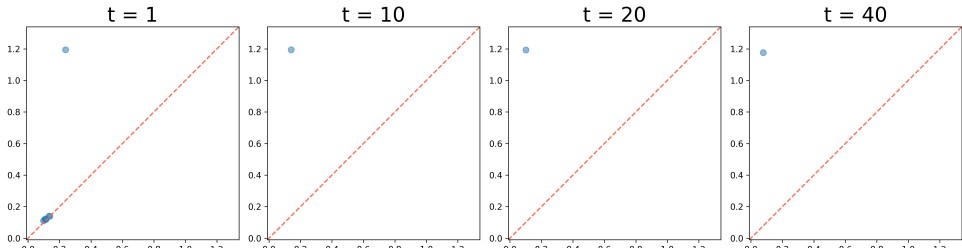

Figure 8: Circle denoising — 1st persistence diagram evolution (combined stages).

## B.2 CIRCLE EMERGING

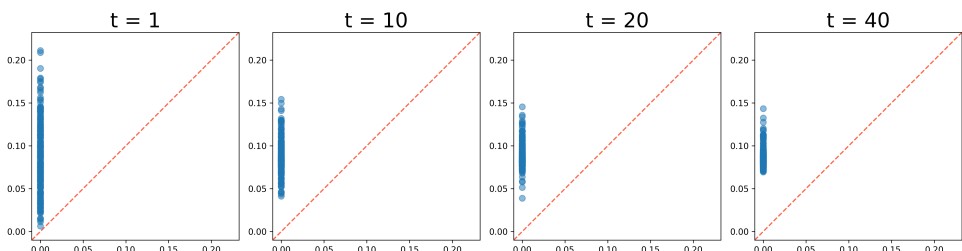

Figure 9: Circle emerging — 0th persistence diagram evolution (combined stages).

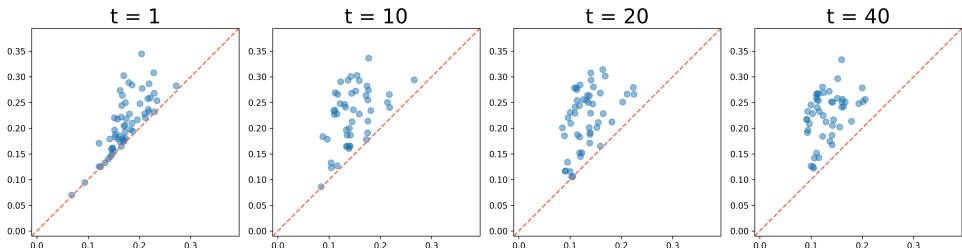

Figure 10: Circle emerging — 1st persistence diagram evolution (combined stages).

