# OpenReview forum: "Dynamical Persistent Homology via Wasserstein Gradient Flow"
_ICLR.cc/2026/Conference — ICLR 2026 Conference Withdrawn Submission_

### Official Review · Reviewer_7Eie · 2025-10-27

**Soundness:** 3
**Presentation:** 2
**Contribution:** 2
**Rating:** 2
**Confidence:** 4

**Summary:**

The paper proposes a way to optimize persistence diagram when the target diagram is not given. To this end, the paper employs the JKO scheme from the optimal transport theory.
The paper also proposes an algorithm to optimize persistence diagrams with the given target diagram. This method uses interpolation in the Wasserstein space.
The paper does not contain a clear list of contributions; from the reviewer's viewpoint, only the first item mentioned above can count as such (Algorithm 2 in the paper).

**Strengths:**

- The paper covers the background from both optimal transport and TDA sides, which is a challenge in itself.
- Overall, the paper is well-structured.

**Weaknesses:**

1. Lack of originality
2. Missing motivation for Alg. 1
3. (assuming the answer to q. 1 is yes) suspiciously non-standard treatment of the Wasserstein distance between PDs as optimal transport problem.

1. The cited paper by Nigmetov and Morozov (by the way, as far as I know, the idea of modifying the shape of input by backpropagating through persistence diagrams first appeared in "A topology layer for machine learning" by Brüel-Gabrielsson et al., which probably should be cited) contains the matching loss as the main motivating example. Using the matching that realizes Wasserstein distance is a natural choice in some (but not all -- the paragraph below) scenarios.

Another point: the authors can mention in the conclusion a limitation of optimal transport approach: it is insensitive to spatial location of the topological features. If I had two holes far apart from each other, they can distorted so that the birth/death values for the deformed holes are such that Wasserstein  matching between the distorted version and the original matches spatially distant holes. Persistence diagram does not tell you anything about the location of your features in the input.

2. I don't understand the need for Alg.1. If I naively use the critical set method for the target diagram that is given to me, I am trying to move the diagram to it directly. If I am using Alg. 1, I am computing some intermediate version and moving the diagram to this version -- but the gradients point exactly in the same direction. The paper needs to have at least enough experimental evidence that introducing this interpolated target buys us anything --- compare the convergence speed and the results of Alg. 1 with the "matching loss for Wasserstein matching".

3. The paper seems to make it very important for the diagrams to have equal cardinality. The standard way of doing that is to add diagonal projections. Instead, the paper uses the 1/n weights which are quite suspicious and require careful explanation and motivation.
By the way, an interesting variation of the experiments for the paper setting could be as follows. Instead of trying to convert an optimal plan to a target diagram, use it directly. The plan wants 0.6 of the mass of the current point (b, d) to go to (b_1, d_1) and 0.4 of its mass to go to (b_2, d_2) --- let's just have these (contradicting) terms in the loss, say, for birth: 0.6 * (1/n) * (b - b_1)^2 + 0.4 * (1/n) * (b - b_2)^2. I assume that the mass of (b, d) is (1/n).

**Questions:**

1. When we compute the optimal matching (realizing the Wasserstein distance) between two persistence diagrams, it is crucial to be able to match points to the diagonal. The text in lines 259-268 and 347 suggests that the paper uses a different approach, considering exclusively off-diagonal points. Is this true?
2. In Alg. 2, is the following interpretation of lines 4, 6, 7, 8 correct: we initialize measure \mu with measure \mu_k, then we solve the argmin in line 7 and the variables y_k are read off from it? It is a bit hard to parse, since the assignment in line 6 does not necessarily imply to a reader that further changes made to \mu will be reflected in y_k-s. Also,  what is the guarantee that we get a persistence diagram (sum of Dirac measures)?

---

### Official Review · Reviewer_nrbU · 2025-10-28

**Soundness:** 3
**Presentation:** 2
**Contribution:** 3
**Rating:** 6
**Confidence:** 3

**Summary:**

The paper introduces dynamical persistent homology: evolve persistence diagrams along Wasserstein gradient flows and pull these dynamics back with differentiable persistent homology to update the filtration or data, reversing the usual TDA direction. It offers two methods, McCann interpolation toward a target via optimal transport and a JKO energy-based scheme without a target, demonstrated on circle denoising and loop emergence.

**Strengths:**

1. Novelty. The paper frames an inverse problem for TDA by evolving persistence diagrams in Wasserstein space and pulling those dynamics back through differentiable persistent homology, bridging optimal transport and persistence optimization.

2. It provides two complementary realizations: McCann interpolation toward a single target diagram with OT couplings and filtration updates, and a JKO energy-based scheme without a target, with clear algorithmic details and discussion of achievability of intermediate steps.

3. Clear, instructive demos. The circle denoising and circle emerging case studies make the approach tangible, including an explicit repulsion loss, concrete energy functionals, and implementation notes like using Sinkhorn and sliced Wasserstein for efficiency.

**Weaknesses:**

1. While the idea is very nice, empirical evidence is limited to qualitative toy demos with circles, without quantitative metrics, baselines, or ablations to assess robustness or superiority.

2. No guarantee that each desired diagram step is realizable by filtration updates; the method explicitly carries forward unattained targets ($Y^{(k)}$), highlighting an achievability gap.

3. Scalability and scope are constrained: current implementation is CPU bound, focuses only on standard persistence diagrams, and lacks statistical characterization or guidance for choosing energies.

**Questions:**

1. Could you add *quantitative evaluations and baselines*, like bottleneck or ($W_p$) distances per iteration, persistence of the principal feature, and ablations over Sinkhorn regularization, JKO step size ($\tau$), and the repulsion weight, to substantiate the toy demos beyond figures?

2. What *conditions or bounds* can you provide to ensure that each desired step ($Y^{(k)}$) along the McCann or JKO evolution is *realizable by a filtration update*, or at least to quantify the mismatch when it is not?

3. If we restrict to point clouds with Rips filtrations, under what conditions can your procedure "realize" a given target persistence diagram? Can you provide a certificate or bound on the mismatch when a JKO or McCann step ($Y^{(k)}$) is infeasible?

4. This question is somewhat tangential to the paper’s main goal but could point to an interesting extension. Suppose we have *time-indexed persistence diagrams* (${D}_{t=1}^T$) from real data. Can your framework *fit a Wasserstein-gradient-flow path* that interpolates these diagrams, and under what assumptions on continuity, cardinality changes, and noise would this be identifiable and stable? For example, how would you choose the energy (J) and JKO step size, and handle unmatched mass near the diagonal across time?

---

### Official Review · Reviewer_HRu9 · 2025-10-30

**Soundness:** 3
**Presentation:** 3
**Contribution:** 2
**Rating:** 4
**Confidence:** 4

**Summary:**

This paper presents two algorithms to approximate (constant-speed) geodesic in
the space of persistence diagrams.
The first one directly computes the interpolation between two given diagrams,
and the second allows shooting a geodesic minimizing an energy functional.

**Strengths:**

- Mathematically sound, and conceptually simple.
 - Code is available, experiments are reproducible via simple notebooks.

**Weaknesses:**

- Theoretical guarantees. The (continuous) theory relies on well established
 optimal transport theory, but the resulting geodesics are discretized. Are
 there some bounds on the error?
 - Experiements.
   - Synthetic datasets only.
   - Can these results be compared to other methods (with other losses)?
   - The experiements are on very small datsets, how does this scale to larger datsets?
   - This approach is based on several approximations schemes. Are there
   empirical validations on known datasets?
   - Are there examples of machine learning on which Algo 1 or 2 is helpful?
 - The definition of the first variation is a bit vague. Is it following from a
 Riesz representation theorem? Is $\xi$ absolutely continuous w.r.t. $\mu$?

minor:
 - l145. $\mathcal P_2$ is defined in the next paragraph
 - l191. "other methods" the [carrière et al] ?
 - l201. Persistence diagrams, homology  and essential features are not introduced.
 - l258. Sinkhorn is not an exact algorithm
 - Reproducibility. I got trouble reproducing the notebooks.
 (module 'oineus' has no attribute 'get_vr_filtration_and_critical_edges').
 The function exists in the main repo, so this might be an issue on my end.

**Questions:**

See weaknesses.
 - In Algo 2. Do we have $Y=X$ ? IIC, the next "target diagram" is set to
 $\mu$, does it update the value of $Y$ there?
 In that case the use of Sinkhorn may be a bit problematic since (IIRC) the
 transport plan given by this algo will not be a permutation (as opposed to its
 non-regularized counterpart). Do you have to use a trick such as `TargetDgm`?
 - As mentionned in the limitations (2), this could be extended to other setups.
 Is there a limitation to not apply it to diagrams coming from zigzag, extended
 persistence, or signed barcodes (in multiparameter persistence)? As far as I
 understand it this extension should be straightforward, since they all behave
 similarly.

---

### Official Review · Reviewer_EvbA · 2025-10-31

**Soundness:** 2
**Presentation:** 2
**Contribution:** 2
**Rating:** 2
**Confidence:** 3

**Summary:**

This paper focuses on the inverse operation of persistent homology—namely, reflecting modifications in persistence diagrams back onto the original data. As an inverse operation method applicable to the recently popular _p_-Wasserstein distance used for comparing persistence diagrams, the authors propose a computational algorithm that leverages McCann interpolation and an energy functional. The convergence of the proposed algorithm is theoretically proven, and its empirical behavior is demonstrated through experiments.

**Strengths:**

This paper proposes an algorithm for solving the inverse problem of persistent homology based on the _p_-Wasserstein distance and provides a proof of its convergence.

**Weaknesses:**

The concerns about this paper can be summarized in two main points:

- First, it is unclear whether the content of this paper fits within the scope of the conference. This venue focuses on AI, machine learning, and representation learning, and the paper should therefore demonstrate its relevance or effectiveness in these contexts. While the proposed method may indeed have potential applications in AI or machine learning tasks, the current presentation does not explicitly describe such connections. As it stands, the work appears to belong more to the domain of computational geometry or mathematical analysis rather than to AI or machine learning.

- Second, the main claim of the paper is not sufficiently clear. Presumably, the authors aim to claim novelty in providing the first convergence-guaranteed algorithm for solving the inverse problem based on the _p_-Wasserstein distance. However, the paper should include comparative experiments illustrating what phenomena occur when convergence is not achieved and how the proposed method resolves those issues. Such comparisons would make the contribution and significance of the proposed approach much clearer.

**Questions:**

Please clarify the contribution of the proposed method to challenges in the field of AI and machine learning, and provide concrete evidence supporting this contribution. In particular, it would be helpful to explain how the proposed approach advances or benefits AI/ML tasks, beyond its theoretical significance in the context of computational geometry or topological data analysis.

---

### Note · Authors · 2025-11-27

I have read and agree with the venue's withdrawal policy on behalf of myself and my co-authors.